# High-Q-Factor Silica-Based Racetrack Microring Resonators

**Yue-Xin Yin** [1], **Xiao-Jie Yin** [2,3], **Xiao-Pei Zhang** [4], **Guan-Wen Yan** [1], **Yue Wang** [2], **Yuan-Da Wu** [2], **Jun-Ming An** [2], **Liang-Liang Wang** [2] and **Da-Ming Zhang** [1,*]

1 State Key Laboratory of Integrated Optoelectronics, College of Electronic Science and Engineering, Jilin University, 2699# Qianjin Street, Changchun 130012, China; yxyin20@mails.jlu.edu.cn (Y.-X.Y.); gwyan19@mails.jlu.edu.cn (G.-W.Y.)
2 State Key Laboratory of Integrated Optoelectronics, Institute of Semiconductors, Chinese Academy of Sciences, Beijing 100083, China; yinxiaojie@semi.ac.cn (X.-J.Y.); wy1022@semi.ac.cn (Y.W.); wuyuanda@semi.ac.cn (Y.-D.W.); junming@semi.ac.cn (J.-M.A.); wangliangl09@semi.ac.cn (L.-L.W.)
3 Shijia Photons Technology, Hebi 458030, China
4 National Laboratory of Solid State Microstructures and College of Engineering and Applied Sciences, Nanjing University, Nanjing 210093, China; DZ1634007@small.nju.edu.cn
* Correspondence: zhangdm@jlu.edu.cn

**Abstract:** In this paper, ultrahigh-Q factor racetrack microring resonators (MRRs) are demonstrated based on silica planar lightwave circuits (PLCs) platform. A loaded ultrahigh-Q factor $Q_{load}$ of $1.83 \times 10^6$ is obtained. The MRRs are packaged with fiber-to-fiber loss of ~5 dB. A notch depth of 3 dB and ~137 pm FSR are observed. These MRRs show great potential in optical communications as filters. Moreover, the devices are suitable used in monolithic integration and hybrid integration with other devices, especially in external cavity lasers (ECLs) to realize ultranarrow linewidths.

**Keywords:** optical devices; resonators; integrated optics; wavelength filtering devices

## 1. Introduction

With the development of ultrahigh-speed optical interconnection, coherent optical communication [1–3], and coherent detection technology [4–7], more urgent requirements are put forward for the narrow linewidth, high power, and high stability of laser source. However, limited to the length of cavity, the spectral linewidth conventional distributed feedback (DFB) lasers and distributed bragg reflector (DBR) lasers are typically in the order of ~MHz, along with a small tuning range around a few nm [8,9]. Therefore, external cavity lasers (ECLs) become the first choice to replace traditional lasers to supply narrow linewidth of ~sub 100 kHz and wide tuning range. The ECLs based on blazed diffraction gratings [4] are of wide tuning range and narrow linewidth, but suffer from huge size, complex package, and difficult alignment, which increase the cost and harm the stability of the lasers. On the other hand, hybrid integration between photonics chips and semiconductor lasers provides an ideal method to fabricated ECLs. These hybrid integration ECLs have been investigated based on different materials, such as III-V [3], silicon [10–12], silica [6], SiN [13], and polymers [14–16]. Compared with other materials, silica planar lightwave circuits (PLCs) are attractive for ECLs owing to its high mechanical strength, high thermal stability, low loss, and almost the same core geometry with fiber cores, which is suitable for an optical communication system. Besides, silica-based waveguides are widely used to fabricate commercial applications, such as optical splitters [17], array waveguide grating [18], and optical buffer [19]. Therefore, silica based ELCs are promising to realize large scale integration for optical communication.

In particular, the microring resonators (MRRs) with compact sizes and ultrahigh-Q factors show great potential in lasers [10–14,20], sensors [21–24], and filters [25–28]. Compared with basic MRRs, the racetrack MRRs using lateral coupling, increase the coupling gap size and the fabrication tolerance. Here, in this paper, ultrahigh-Q factor

racetrack MRRs are designed, fabricated, and experimentally demonstrated. The proposed racetrack MRRs are built on silica PLCs platform. Owing to the ultralow loss of the waveguide, ultrahigh Q factors of $1.83 \times 10^6$ are acquired. From measured transmission, a notch depth of 3 dB and ~137 pm FSR is observed. The MRRs are packaged with fiber-to-fiber loss of ~5 dB. These MRRs are suitable used in optical communications as filters. Moreover, the devices are potential used in monolithic integration and hybrid integration with other devices, especially used in external cavity lasers (ECLs) to realize ultra-narrow linewidths.

## 2. Device Design and Simulation

The MRR proposed is fabricated on a silica-based PLC platform. The cross-section of the silica waveguide is shown in the inset of Figure 1a. In order to realize compact footprint, the refractive index difference between the core and cladding is 2% with different doping. The refractive indices of the cladding and the core are 1.4448 and 1.4737(@1550 nm) respectively. The single-mode condition is calculated by MATLAB based on the eigenvalue equations [29], and the thickness of the core is selected to be 4 μm according to the simulation results in Figure 1a. As Figure 1b shows, the mode profile of the fundamental transverse electric (TE) mode is simulated through beam propagation method (BPM) and the effective refractive index is 1.4615.

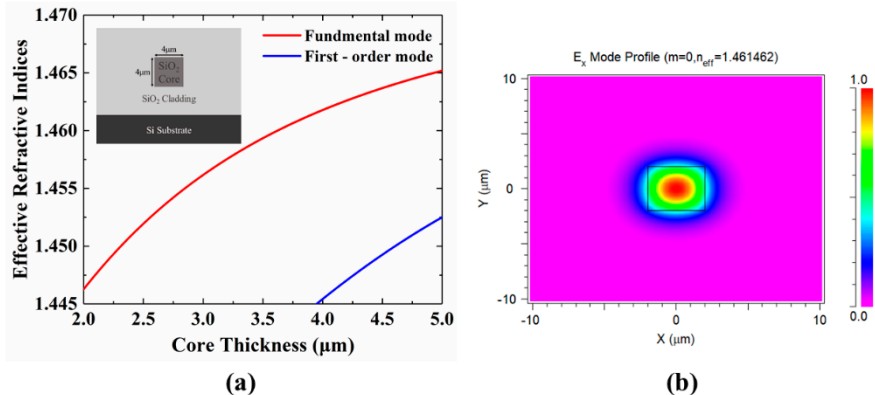

**Figure 1.** (**a**) Relations between the core thickness and the effective refractive indices at 1550 nm. The inset shows the cross-section of waveguide. (**b**) The mode profile of the waveguide simulated through BPM method.

The structure of racetrack MRRs with directional couplers (DCs) is shown in Figure 2. Using the coupled mode theory (CMT) and the transfer matrix technique (TMT) [30,31], we derive the transmission power $P_{t1}$ in the output waveguide, which is,

$$P_{t1} = |E_{t1}|^2 = \frac{\alpha^2 + |t|^2 - 2\alpha|t|\cos(\theta)}{1 + \alpha^2|t|^2 - 2\alpha|t|\cos(\theta)} \tag{1}$$

where $\alpha$ is the loss coefficient of the ring, and $t$ and $\kappa$ are the coupler parameters with the relation of $|\kappa^2| + |t^2| = 1$. The phase shift in the ring is,

$$\theta = L_{cir}\beta \tag{2}$$

where $L_{cir}$ is the circumference of the racetrack which is given by $L_{cir} = 2\pi R + 2L$, R being the radius of the ring measured from the center of the ring to the center of the waveguide, L being the coupling length. The propagation constant $\beta$ is,

$$\beta = \frac{2\pi}{\lambda} \cdot n_{eff} \tag{3}$$

where $\lambda$ is the wavelength, $n_{eff}$ is the effective refractive index.

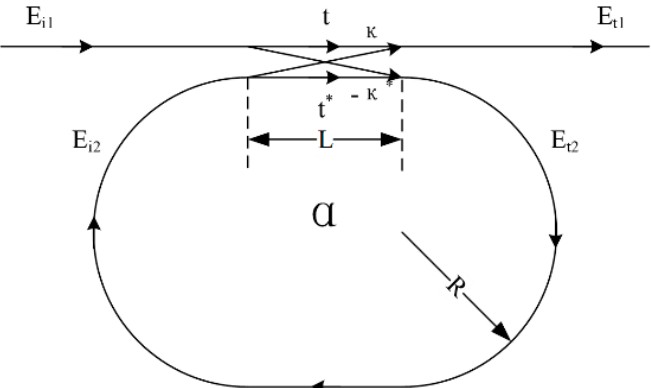

**Figure 2.** Schematic of the racetrack micro-ring resonator.

Figure 3 shows the relations between transmission efficiency and bend radius. While the bend radius is larger than 1460 μm, bend loss could be lower than 1 dB/cm. From the resonance equation of MRRs, we can derive the expression of the ring radius $R$ and the free spectral range (FSR) as

$$R = \frac{m\lambda_0}{2\pi n_{eff}}, FSR = \frac{\lambda_0 n_{eff}}{m n_g} \tag{4}$$

where $m$ is the resonator order of the MRRs, $\lambda_0$ is the central resonant wavelength, $n_{eff}$ is the effective refractive index, and $n_g = n_{eff} - \lambda dn_{eff}/d\lambda$ is the group refractive index. According to Figure 4, with a decrease in $R$, $FSR$ will increase dramatically. However, the bend loss will also increase. Low bend loss means high Q factor, but small $FSR$ will limit the applications—for example, the port number of wavelength division multiplex (WDM) systems and working range of sensors. Taking the balance between these parameters, we select $m$ to be 9519. In this case, from Equation (4) we can estimate the ring radius $R$ is 1600 μm, and $FSR$ to be 161 pm. With such ring radius, the bend loss of 0.83 dB/cm is calculated.

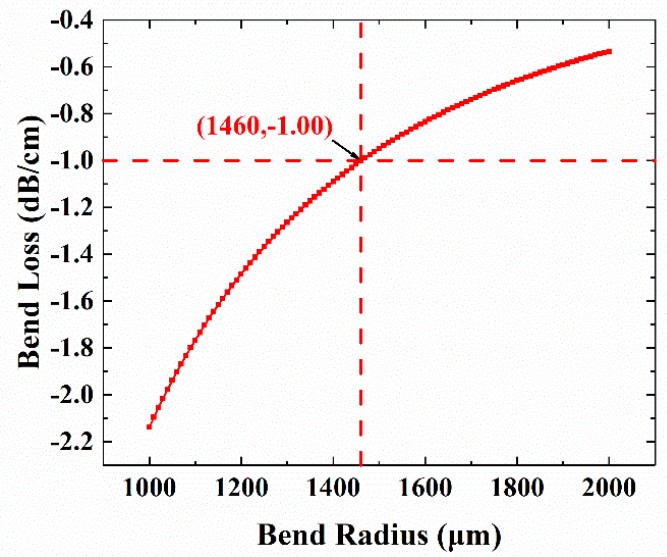

**Figure 3.** Relations between bend-loss and bend-radius.

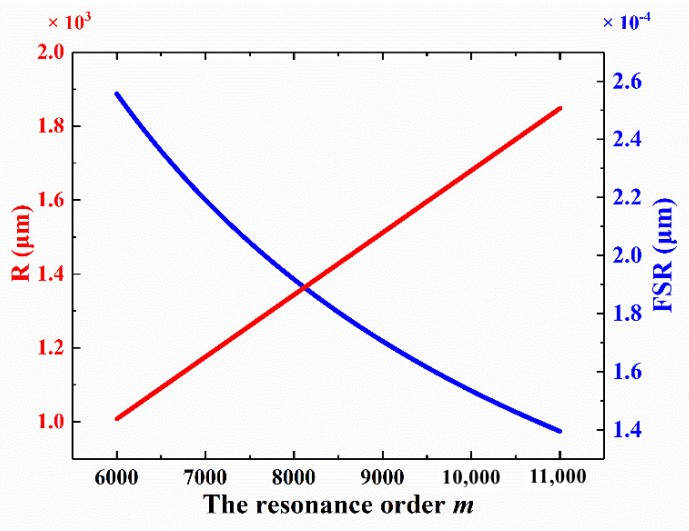

**Figure 4.** Relations between the resonance order m and the ring radius R and the free spectral range (FSR).

According to typical loss of silica-based waveguide we published [19], we choose the loss of waveguide is 0.1 dB/cm to optimize the parameter of the racetrack MRR. The loss is an average loss of curved and straight waveguides. Then the loss coefficient $\alpha$ (zero loss $\alpha = 1$) is 0.984 of the ring with 1600 μm radius and 800 μm coupler length. Based on Equation (1), we calculate the coupling conditions of the MRR through MATLAB. As Figure 5 shows, the calculate results indicated the deepest notch, which is the closest to the critical coupling condition, is obtained while the coupling coefficient $\kappa$ is 0.2. Then, we choose the gap of DCs is 4 μm to realize the coupling ratio $|\kappa^2|$ of about 4.4%. Finally, the simulated transmission spectrum is obtained. As Figure 6a shows, ~141 pm FSR and ~18 dB depth is acquired. Figure 6b shows one peak around 1546.305 nm. The simulated data shown by the blue hollow circles were fit using the theoretical Lorentzian transmission in Origin (see the red solid curve). It can be seen that the full width at half maximum (FWHM) of the resonance peak for the present resonator is about $\Delta\lambda = 0.656$ pm, which indicates that a simulated Q factor $Q_{sim}$ of $2.36 \times 10^6$ is obtained.

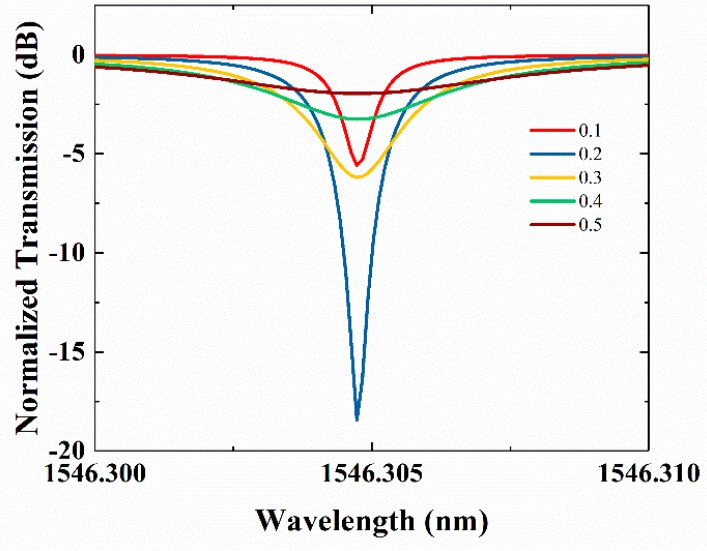

**Figure 5.** Relations between coupling efficiency and dip depth.

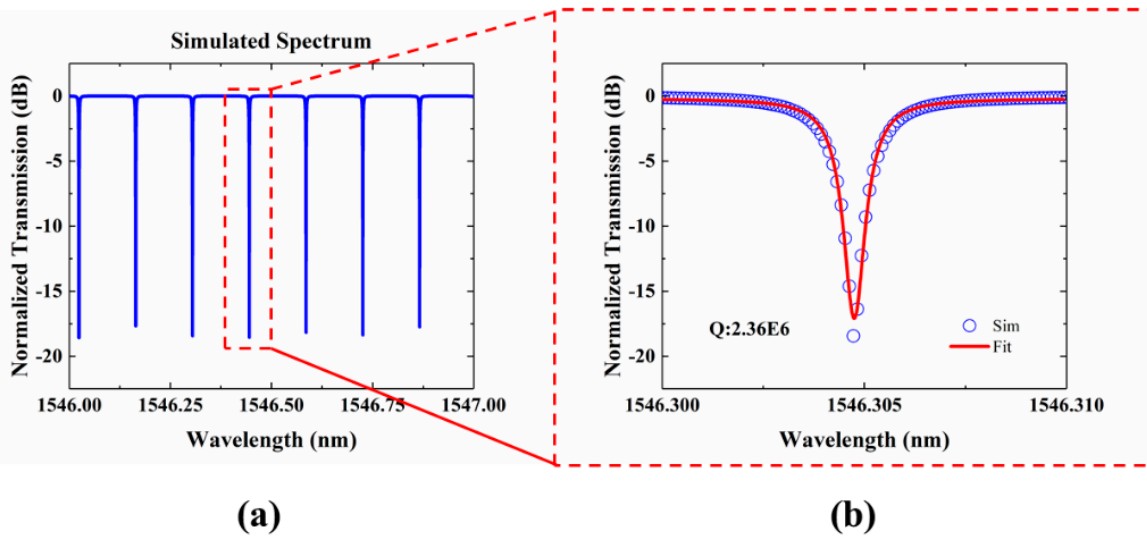

**Figure 6.** (**a**) Part of the simulated transmission spectrum. (**b**) One peak of simulated transmission spectrum with Lorentzian fitting.

## 3. Device Fabrication and Characterization

The designed MRR is fabricated by the PLC foundry, SHIJIA, China on a 6 inch silica-on-silicon wafer. First of all, the bottom cladding silica layer of ~15 μm is thermally oxidized on silicon substrate. Then, the core-layer is deposited by plasma-enhanced chemical vapor deposition (PECVD). After, the core layer is annealed above 1100 °C to be compact. We pattern the waveguides through UV lithography and fully etch the core layer by inductively coupled plasma (ICP) dry etching process with a $C_4F_8/SF_6$ gas mixture. Next, the top cladding of ~15 μm is deposited by PECVD and annealed again. Figure 7a shows the picture of the proposed MRRs coupled with fiber arrays (FAs). Figure 7b shows the cross section of the core waveguide. However, the geometry of core is not exactly a square due to the fabricating process, leading to transverse magnetic (TM) mode exists in waveguides. The device is designed to be working under TE mode, as its notch depth is much larger than that TM mode. We can distinguish different mode in the same spectrum easily.

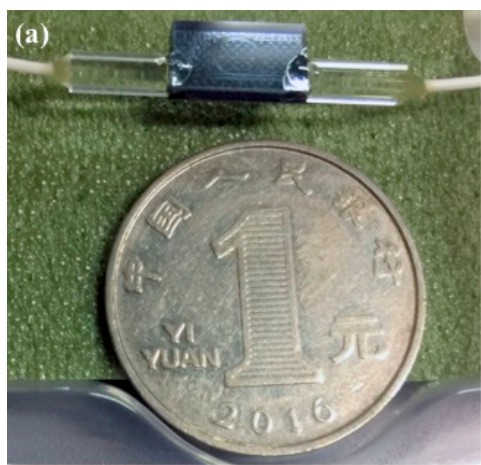
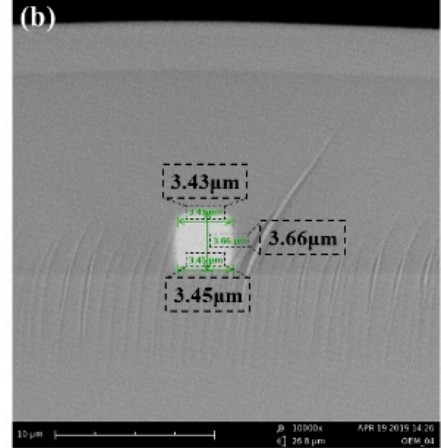

**Figure 7.** (**a**) Microring resonator (MRR) after package process. (**b**) The scanning electron microscope (SEM) photo of the core waveguide cross section

To measure the fabricated optical structures, we use a swept-wavelength laser source (New Focus TLB-6600) that is tunable from 1510 to 1590 nm. The output signal of the MRR is detected by a photodetector (Thorlabs PDA10CS, 17 MHz bandwidth) associated with a

data-acquisition card (DAQ, National Instruments USB-6366). The measured transmissions were normalized with respect to the transmission from laser source to photodetector. Owing to the present resonator having an ultra-high Q-factor, the step size of the tunable laser source was set to be 0.03 pm. Figure 8a shows part of the transmission spectrum. From this picture, the TE mode and the TM mode can be divided obviously. Besides, FSR of ~137 pm is also observed, which is consistent with the theoretical analysis and the simulation result. However, the notch depth of ~3 dB is much lower than simulation. The reason might be the bend coupling increase the coupling coefficient. The MRRs are well packaged with fiber-to-fiber loss of ~5 dB and can be used directly. As Figure 8b shows, the measured data shown by the blue hollow circles were fit using the theoretical Lorentzian transmission in Origin (see the red solid curve). It can be seen that the full width at half maximum (FWHM) of the resonance peak for the present resonator is about $\Delta\lambda = 0.839$ pm, which indicates that a loaded ultrahigh-Q factor $Q_{load}$ of $1.83 \times 10^6$ is obtained. The $Q_{load}$ is lower than $Q_{sim}$ owing the over-coupling situation.

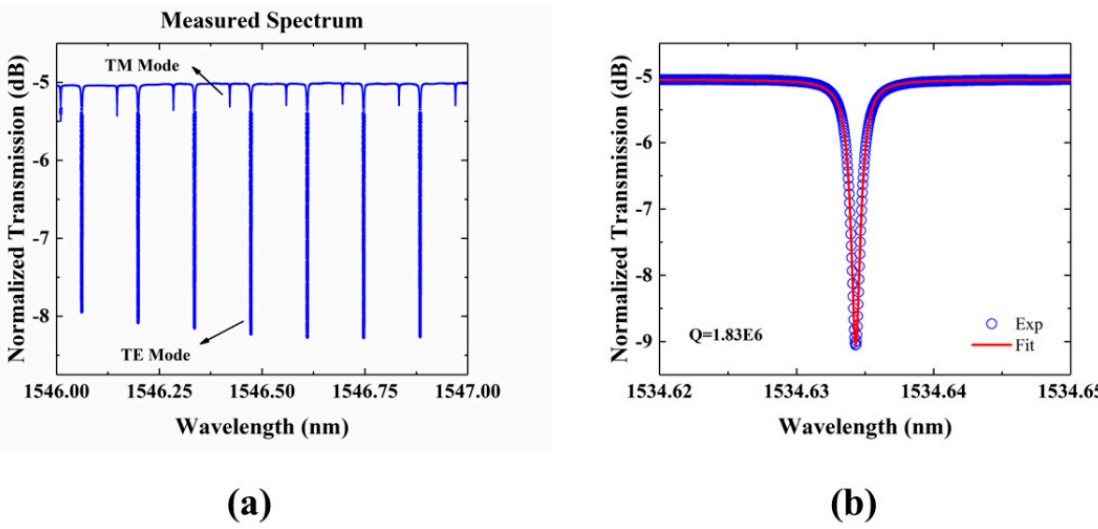

**Figure 8.** (**a**) Part of the measured transmission spectrum. (**b**) One peak of measured transmission spectrum with Lorentzian fitting.

## 4. Discussion

Table 1 shows a comparison of the microring resonators reported in the recent years that have a Q-factor higher than $10^5$ based on different platforms. In order to increase the Q-factor of MRRs, fabrication process and structures have been optimized. In Reference [32], an intrinsic Q factors of $7.60 \times 10^5$ is acquired through etch-less fabrication processes with thermal oxidation. However, the thermal oxidation process is not standard for silicon photonics and unavailable for foundries. The same problem is met in Reference [33]. Optimized ICP and chemical mechanical polishing (CMP) are utilized to reduce sidewall and surface roughness. Though $3.70 \times 10^7$ Q-factor is acquired, the method is intended for SiN waveguides. On the other hand, bend couplers [34] and large radius [35] are used. In Reference [34], the Euler bend couplers with the effective radius R of 29 μm achieve a Q-factor as high as $1.30 \times 10^6$ and 0.9 nm FSR. However, the design shows lack of tolerance. As for [35], 6000 μm radius decrease the loss but also FSR to 0.017 nm, which limits applications of the devices. Owing to the loss of polymer waveguides [21], it is hard to get a Q-factor of $>10^6$. In Reference [20], a rolled-up microtubes (RUMs)/ silicon-on-insulator (SOI) waveguide system with high-quality-factor of $1.5 \times 10^5$ is demonstrated. Once the transfer process is simplified, the system will be used more widely in the field of optical communications. The proposed silica-based racetrack MRRs are fabricated by the PLC foundry on a 6 inch silica-on-silicon wafer toward commercial use. The fabrication process insures ultralow loss waveguide. Hence, an ultrahigh Q-factor of $1.83 \times 10^6$ is acquired.

Besides, the devices are suitable used in monolithic integration and hybrid integration with other devices, especially used in ECLs to realize ultra-narrow linewidths.

**Table 1.** Comparison of Ultrahigh-Q Microring Resonators.

| Ref. | Platform | R (μm) | FSR (nm) | Q-Factor |
|------|----------|--------|----------|----------|
| [32] | silicon | 50 | N.A. | $7.60 \times 10^5$ |
| [34] | silicon | 29 | 0.900 | $1.30 \times 10^6$ |
| [35] | silicon | 6000 | 0.017 | $1.70 \times 10^6$ |
| [33] | SiN | 115 | N.A. | $3.70 \times 10^7$ |
| [21] | Polymer planar lightwave circuits (PLC) | 60 | 2 | $8.00 \times 10^5$ |
| [20] | Microtube/SOI | 3.5 | 3~33 | $1.5 \times 10^5$ |
| This Work | Silica PLC | 1600 | 0.137 | $1.83 \times 10^6$ |

## 5. Conclusions

In summary, we demonstrate ultrahigh-Q factor racetrack MRRs based on silica PLCs platform. A loaded ultrahigh-Q factor $Q_{load}$ of $1.83 \times 10^6$ is obtained. A notch depth of 3 dB and ~137 pm FSR are observed. The measured spectrum is consistent with the theoretical analysis and the simulation result. The MRRs are well packaged with fiber-to-fiber loss of ~5 dB and can be used directly. The Q factor and depth of notches can be improved by tuning to critical coupling situation through tunable MRRs structure. The FSR can also be enlarged through cascade high-order MRRs. These MRRs show great potential in optical communications as filters. Moreover, the devices are suitable used in monolithic integration and hybrid integration with other devices.

**Author Contributions:** Methodology, Y.-D.W.; software, Y.-X.Y.; validation, Y.W., J.-M.A. and L.-L.W.; formal analysis, X.-P.Z.; investigation, Y.-X.Y. and G.-W.Y.; resources, Y.W., J.-M.A. and L.-L.W.; data curation, Y.-X.Y. and G.-W.Y.; writing—original draft preparation, G.-W.Y.; writing—review and editing, Y.-X.Y.; project administration, D.-M.Z.; funding acquisition, D.-M.Z. and X.-J.Y. All authors have read and agreed to the published version of the manuscript.

**Funding:** This research was funded by National Key Research and Development (R&D) Program of China (2019YFB2203004) and Science and Technology Development Plan of Jilin Province (20190302010GX).

**Institutional Review Board Statement:** Not applicable.

**Informed Consent Statement:** Not applicable.

**Data Availability Statement:** Not applicable.

**Conflicts of Interest:** The authors declare no conflict of interest.

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
