# Peer review of "High-Q-Factor Silica-Based Racetrack Microring Resonators"

_photonics, doi:10.3390/photonics8020043_

Round 1

Reviewer 1 Report

in this work, authors designed and fabricated high quality cavities based on SiO2 with a law loss.They showed a nice comparison of the microring resonators which were fabricated in the recent years. However they did not talked/reported different types of resonators. for example, rolled-up microtube resonators whichare novel type of resonators and new member of this big family. I really asked authors to consider this discussion as well. for more details you can see and cite this paper:

(1) https://www.osapublishing.org/ol/abstract.cfm?uri=ol-39-2-189

other parts of the paper are looking okay and I can consider this work for publication after authors applied all of my comments and concerns.

Reviewer 2 Report

The authors presented a racetrack mirroring resonator made of silica. the design of the resonator lacks some in depth explanation such as eq used for the Matlab model, boundary conditions, and so forth. the fabrication used is classical lithography; however, the authors claim that this resonator is potential for "manufacture". Not sure what that sentence is referred to and where the novelty is here. the work is similar to others (see for example "Pauline Girault, Nathalie Lorrain, Jonathan Lemaitre, Luiz Poffo, Mohammed Guendouz, et al.. Race- track micro-resonators based on ridge waveguides made of porous silica. Optical Materials, Elsevier, 2015, 50 (part B), pp.167-174. 10.1016/j.optmat.2015.10.017 . hal-01588445"). The authors should explain where the novelty here is and also enrich the literature review part with more papers related to Racetrack Microring Resonators. some English edits needed (see attached with highlights) and the use of some better wording.

Reviewer 3 Report

This Paper "High-Q-Factor Silica-Based Racetrack Microring Resonators" reports interesting results in terms of quality factor of a silica based optical microring resonator fabricated on Si substrate and designed to work at telecommunication wavelengths (1.5μm). Such values of Q-factors (≈106) should be of great interest, for instance to filtering effect.

In my humble opinion, Although the results reported are interesting, the paper is not very well organized and the some relevant points are not understandable. The paper needs major revisions to be published on MDPI.

As my suggestions of modification of the manuscript to the authors, please consider the following points:

1 - About the form: the manuscript version is, without doubt, not the last one because there are to much mistakes as:

  • the mixing between red and black police
  • Line 19, "What's more" seems to be not very appropriated sentence for a publication
  • just before the keywords a sentence not deleted (lines 22 and 23),
  • from line 47 to 54, it's the exact copy of 50% of the abstract
  • line 46, "10^6"
  • line 59 and 60: put the plural to indices = the refractive indices
  • Figure 2 is not at the right scale
  • Legend of the figure 3, mistake with the word "resonance"
  • End of Line 115: Supress 1600
  • line 120: Mistake with the word "Discussion"
  • Table 1: two lines with a mistake of two values for the same study
  • Maybe other faults, need to be finely verify

2 - About the content:

  • line 61: explain the calculation method and eventually the references used.
  • Figure 1:
    a - dimension of the SiO2 Cladding and distance between Si substrate and Core? Shape of the mode? Gamma Core (ratio between the part of the light in the Core and in the cladding)
    b - Ey ? which axis is defined as y? The superposition of E10 and E01 is not enough clear
  • Line 70: Why 5%? You have to tell more and justify your choice about such coupling conditions, we know that in undercoupling conditions, the quality factor is the best but why this value?
  • Line 72, equation 1: I don't understand why it's interesting to explain the well known R and FSR in a such study. 
  • Figure 3: You have probably to satisfy a compromise between two parameters, so please, explain it in order to understand the role of this figure
  • Line 83: the bending losses are neglected for R=1600μm, maybe, but you have to prove it relying on previous work or references in the similar situation.
  • Lines 83-84: Not understandable, it's the first time in the manuscript the quality factor is addressed...
  • Sentence line 106: Nothing before in the document explain Q 
  • Line 111: Give the model using Lorentz method
  • Sentence line 114-115: 2.14 10^6 is incomprehensible because an another value just before is given.
  • Line 115, how did you extract the propagation losses, from the Lorentzian fit? Please give more informations.
  • Figure 6, why 4 resonances? An other possibility should be only one resonance plotted and in the text the explanation of the statistics for more peaks near the presented one. The wider spectrum in figure 5 could be centered with the focused peak

Finally, I think the organization of the study have to be reorganized for a better understanding of the choice of the design and the links between the R, the Gap, the losses and finally the Q-factor.

Round 2

Reviewer 1 Report

With this current form and results, I am happy to say that the revised paper can be consider for the publication.  

Author Response

Thank you very much for your comments and valuable suggestions.

Yue-Xin Yin, Da-Ming Zhang etc.

Reviewer 2 Report

Thank you for revising the paper

Author Response

(The authors gave the same response as above.)

Reviewer 3 Report

The manuscript improvement is significant, congratulation for that. The figures and explanations added are welcome. I think the article can be published after some comments/suggestions as following :

  • Line 85: A simulation of the transmission coefficient is done for a part of the MRR, why not for a complete round trip? Why not presenting the losses as alpha_bending in dB/cm?
  • Line 92 to 96: The tradeoff between the low losses and FSR is not very well explained, I suggest to modify softly this part focusing the discussion to minimize the looses to reach the highest Quality Factor, a choice is taken to be better than 99% of transmission, OK, but the impact is on the FSR is not understood? Explain the risk in term of application for example.
  • Line 98: Legend 4 is not on the same page of the Figure
  • Line 102: Question: alpha = 0.984 is the sum of the propagation AND bending losses? Not very clear
  • Line 105: mistake: the closest
  • Line 108: What about the Q simulated? It should be a well justification for the fine tunable source and the short step size notified at line 135. An inset within the Figure 6 should be nice with a zoom a one peak.
  • Line 172: The conclusion is too much similar with the abstract, I suggest to give some relevant points about the different ways to investigate in order to increase performances (the contrast for instance)

Author Response

(The authors gave the same response as above.)
